# Organized primary human papillomavirus–based cervical screening: A randomized healthcare policy trial

K. Miriam Elfström[1,2], Carina Eklund[1], Helena Lamin[1], Daniel Öhman[2],
Maria Hortlund[1], Kristina Elfgren[3], Karin Sundström[1], Joakim Dillner[1]*

1 Center for Cervical Cancer Prevention, Karolinska University Laboratory, Karolinska University Hospital and Department of Laboratory Medicine, Karolinska Institutet, Stockholm, Sweden, 2 Regional Cancer Center of Stockholm-Gotland, Cancer Screening Unit, Sweden, 3 Department of Obstetrics and Gynecology, Karolinska University Hospital and Division of Obstetrics and Gynecology, Department of Clinical Science, Intervention and Technology, Karolinska Institutet, Stockholm, Sweden

* joakim.dillner@ki.se

## Abstract

**Data Availability Statement:** The data is deposited with the Swedish Cervical Screening Registry (www.nkcx.se_en) and can be requested at

### Background

Clinical trials in the research setting have demonstrated that primary human papillomavirus (HPV)-based screening results in greater protection against cervical cancer compared with cytology, but evidence from real-life implementation was missing. To evaluate the effectiveness of HPV-based cervical screening within a real-life screening program, the organized, population-based cervical screening program in the capital region of Sweden offered either HPV- or cytology-based screening in a randomized manner through a randomized healthcare policy (RHP).

### Methods and findings

A total of 395,725 women aged 30 to 64 years that were invited for their routine cervical screening visit were randomized without blinding to either cytology-based screening with HPV triage ($n = 183,309$) or HPV-based screening, with cytology triage ($n = 212,416$ women) between September 1, 2014 and September 30, 2016 and follow-up through June 30, 2017. The main outcome was non-inferior detection rate of cervical intraepithelial neoplasia grade 2 or worse (CIN2+). Secondary outcomes included superiority in CIN2+ detection, screening attendance, and referral to histology.

In total, 120,240 had a cervical screening sample on record in the study period in the HPV arm and 99,340 in the cytology arm and were followed for the outcomes of interest. In per-protocol (PP) analyses, the detection rate of CIN2+ was 1.03% (95% confidence interval (CI) 0.98 to 1.10) in the HPV arm and 0.93% (0.87 to 0.99) in the cytology arm ($p$ for non-inferiority <0.0001; odds ratio (OR) 1.11 (95% CI 1.02 to 1.22)). There were 46 cervical cancers detected in the HPV arm (0.04% (0.03 to 0.06)) and 48 cancers detected in the cytology arm (0.05% (0.04 to 0.07)) ($p$ for non-inferiority <0.0001; OR 0.79 (0.53 to 1.18)). Intention-to-screen (ITS) analyses found few differences. In the HPV arm, there was a modestly

info@nkcx.se (administrator Sara Nordqvist Kleppe).

**Funding:** Supported by the Stockholm county healthcare system (www.sll.se) and by a European Union 7th framework program grant to JD (CoheaHr, FP7-F3-2013-603019) (europa.eu/european-union). The funders had no role in study design, data collection and analysis, decision to publish, or preparation of the manuscript.

**Competing interests:** The authors have declared that no competing interests exist.

**Abbreviations:** ASCUS, atypical squamous cells of undetermined significance; CI, confidence interval; CIN1, cervical intraepithelial neoplasia grade 1; CIN2+, cervical intraepithelial neoplasia grade 2 or worse; CIN3, cervical intraepithelial neoplasia grade 3; CONSORT, Consolidated Standards of Reporting Trials; ERB, ethical review board; HPV, human papillomavirus; HSIL+, high-grade squamous intraepithelial lesions or worse; IT, information technology; ITS, intention to screen; LSIL, low-grade squamous intraepithelial lesion; OR, odds ratio; PIN, personal identification number; PP, per protocol; RCT, randomized clinical trial; RHP, randomized healthcare policy.

increased attendance after new invitations (68.56% (68.31 to 68.80) vs. 67.71% (67.43 to 67.98); OR 1.02 (1.00 to 1.03)) and increased rate of referral with completed biopsy (3.89% (3.79 to 4.00) vs. 3.53% (3.42 to 3.65); OR 1.10 (1.05 to 1.15)).

The main limitations of this analysis are that only the baseline results are presented, and there was an imbalance in invitations between the study arms.

## Conclusions

In this study, we observed that a real-life RHP of primary HPV-based screening was acceptable and effective when evaluated against cytology-based screening, as indicated by comparable participation, referral, and detection rates.

## Trial registration

ClinicalTrials.gov NCT01511328

## Author summary

### Why was this study done?

- Evidence from clinical trials has shown that primary human papillomavirus (HPV)-based screening results in greater protection against cervical cancer than cytology-based screening.

- Evidence from real-life implementation of primary HPV-based screening in routine healthcare is scarce.

- We sought to generate evidence on the effectiveness and acceptability of HPV-based screening within an organized screening program.

### What did the researchers do and find?

- Women invited to organized cervical screening in Stockholm, Sweden were randomized to invitation to either primary HPV-based screening or primary cytology-based screening.

- Women were followed for histopathologically confirmed high-grade cervical cancer precursors (cervical intraepithelial neoplasia grade 2 or worse, CIN2+) as the primary outcome. Screening attendance and referral rates to histology were secondary outcomes.

- The detection of CIN2+ was comparable between both screening methods.

- Among women receiving primary HPV-based screening, there was a modestly increased attendance after new invitations and an increased rate of referral with completed biopsy.

### What do these findings mean?

- When organized primary HPV-based screening was implemented in a real-life screening program, it was both acceptable and effective.

- The main limitations of this analysis are that it contains only baseline results and that there was an imbalance in invitations between the 2 screening methods.

## Introduction

Cervical screening is a globally recommended healthcare policy [1]. The human papillomavirus (HPV) is known to be the major causal agent of this cancer [2]. Since 2015, both global [1] and European Union recommendations [3] state that HPV-based cervical screening is the preferred screening modality. In the 2008 EU screening recommendations for cervical cancer, HPV-based screening was classified as an evidence-based screening modality, but implementation was recommended only within randomized healthcare policies (RHPs) [4]. RHP means that 2 different healthcare policies are used in real-life healthcare, where the policy to use is determined by randomization. This strategy is particularly well suited for situations where efficacy and safety are known from studies in the research setting, but where there are uncertainties about how this applies to the real-life healthcare setting. As the policies are randomized, an unbiased quantification of the effects and safety of the policies when used in real-life healthcare can be obtained [5]. The RHP design has been extensively used when implementing new healthcare policies, e.g., for management of low-grade cervical cytological abnormalities [6,7], for treatment choice in prostate cancer [8], and for colorectal cancer screening implementation [9]. To our knowledge, one previous study conducted in Finland has examined the implementation of HPV-based screening using a clearly defined RHP approach [10].

While randomized clinical trials (RCTs) have demonstrated reliable evidence of an improved effect, they are, by definition, performed in the research setting, making generalizability to routine healthcare uncertain. The next step in moving toward broader, routine implementation of HPV-based screening was therefore to examine HPV-based screening using an RHP design in an organized screening program. The purpose of this study was to generate evidence on the effectiveness and acceptability that were inherently generalizable to the real-life setting and yet still benefitted from the temporal comparativeness and reduction of bias achieved through randomization. From January 2012 to May 2014, we started the randomized implementation of HPV-based screening in the greater Stockholm area by targeting women aged 56 to 60 years only [11]. We found that both the major indicator of effectiveness (detection rates of high-grade cervical cancer precursors (cervical intraepithelial neoplasia grade 2 or worse, CIN2+) and of acceptability (population attendances) were similar for both policies/arms [11] and decided to continue with a full-scale randomized implementation targeting all resident women in the ages 30 to 64. As primary HPV screening became the recommended modality in 2015 [12], the RHP was stopped in 2016, and all women aged 30 to 64 are now offered HPV-based screening. The aim of the current study was to evaluate the effectiveness and acceptability of HPV-based cervical screening of the organized program during the time that the program offered either HPV-based or cytology-based screening, in a randomized manner, to about 400,000 resident women.

## Methods

### Study implementation and randomization

During the study period, the cervical cancer screening program invited resident women to screening at 3-year intervals between ages 23 and 50 and 5-year intervals between ages 51 and 64. The upper age limit was extended from 60 years of age to 64 in 2015. Women are invited by letter to screening when 3 or 5 years has passed since their last normal test was taken. The population coverage of being tested is high: 82% of resident women in the target ages were tested according to recommendations in 2015 [13]. However, the population test coverage should not be confused with the invitation attendance rate, which is lower and represents the proportion of women who participate following an invitation. The invitation attendance rate is lower because a high proportion of invitations are renewed invitations that are sent to nonattending women. The following section describes the study implementation and randomization (see also the study protocol and analysis plan, S1 Text).

In August 2014, the full-scale RHP was launched, targeting all resident women ages 30 to 60 and further expanded in January 2015 to include women ages 30 to 64 to reflect changes in the upper age limit. Women invited to screening were randomized to 2 different screening policies: primary cytology with HPV test as triage for women with low-grade cytology (atypical squamous cells of undetermined significance (ASCUS) and cervical intraepithelial neoplasia grade 1 (CIN1)/low-grade squamous intraepithelial lesion (LSIL)) (old policy) or primary HPV-based screening with cytology triage of HPV–positive women (new policy). The randomization was done using coin toss, and arms were assigned based on an even/odd dichotomization of the last control digit in the personal identification number (PIN) of the women. We tested whether a computer randomization of the entire population would give similar results. This was the case, but we preferred to use the randomization using PIN as this prevented changes of randomization status. In accordance with best practice in RHP, the study arm allocation and invitation letters (denoting analysis to be used) were not blinded since RHP follows the principles of clinical practice where the patients are informed about their treatment [5]. Neither the laboratory processing of samples nor the clinical workup of the women were blinded, as we studied the real-life effectiveness of a real program.

Before the RHP was piloted, the protocol was discussed by the regional committee of physicians specialized in gynecology as well as in a national hearing and found to be in accordance with the current scientific evidence. The RHP protocol was approved by the regional ethical review board (ERB) of Stockholm (Decision number 2011/1298-31/3). The ERB decided that the act of participating in screening following an invitation constituted appropriate consent for participation, and further consent collection procedures were not required. The RHP is registered at www.clinicaltrials.gov (registration number NCT01511328). The trial was powered to include >220,000 women to be able to demonstrate non-inferiority in CIN2+ rates (assumed to be 0.9%) between arms at 80% power and 95% confidence level. A non-inferiority margin of 0.10 was assigned as a clinically relevant for this study.

### Sample collection and study procedures

Screening samples were collected by trained midwives at the 63 maternity care units that are contracted as screening stations by the organized program. All samples were taken using the same liquid-based cytology system (ThinPrep, Hologic, Boxborough, MA, United States of America). The samples were collected identically, regardless of which primary analysis method, HPV or cytology that was used.

The HPV testing system was, following an open tender, purchased from Roche Molecular Systems (South Branchburg, New Jersey, USA; Cobas 4800 HPV test) and performed in an accredited laboratory. The Cobas platform tests for HPV types 16, 18, 31, 33, 35, 39, 45, 51, 52, 56, 58, 59, 66 and 68. All HPV positive samples (regardless of HPV type) were subjected to reflex cytology. Samples with invalid results were retested once; if still invalid, a new sample was taken from the woman. Cytology results were communicated using the existing notification letters in the program. The notification letter sent to women with HPV positive and cytology negative results reported that HPV was detected in the sample, but that the cytology was normal.

Women in the HPV policy arm who were high-risk HPV positive and had abnormal cytology were referred by the screening program to colposcopy with biopsy according to established guidelines. HPV–positive women with normal cytology were followed up with a new HPV test after 3 years. If the second HPV test is positive, they will be referred to colposcopy with biopsy (this follow-up will be the focus of forthcoming analysis). In the cytology arm, women with high-grade lesions (high-grade squamous intraepithelial lesions or worse (HSIL+)) in primary cytology were always referred directly to colposcopy with biopsy. Those with ASCUS or CIN1 in cytology were triaged with HPV. If they were HPV positive, they were then referred to colposcopy. Women that had an HPV–negative ASCUS/CIN1 cytology result were invited 3 years later for a new test. The clinical follow-up adhered to established guidelines and was the same in both arms for women referred to colposcopy.

## Data sources and statistical analysis

All resident women that were invited to screening between September 1, 2014 and September 30, 2016 were included. The first invitation in the study period was used as the index invitation, and women were used as the unit of analysis. Information on invitations and test results (HPV, cytology, and histology) was collected from the population-based (i.e., all events in the population are registered) regional screening register. As the randomization used the PIN, it was not possible for individuals to change randomization status. There were no exclusions or losses to follow-up since both the program and registry used were population based. Screening participation (follow-up until December 31, 2016), referrals to colposcopy, and detection rates of cervical abnormalities and cancer (follow-up until June 30, 2017) were estimated by arm and compared overall (intention to screen (ITS), all women randomized) and for organized screening participants separately (per protocol, PP). All women who were sent an invitation and had a sample on record were included in ITS analysis. In the PP analysis, screening participation and sample analysis according to the assigned study arm were also required.

Descriptive statistics were estimated by arm and for the ITS and PP populations. A Farrington–Manning score test for non-inferiority, which is particularly applicable in situations where a materially significant difference or equivalence is of interest in binomial comparative trials, was used to compare the detection rates [14]. Safety of the new policy was defined as non-inferiority in the detection of cancer. A Cochran–Armitage test of trend was used to evaluate changes in the proportion of screen positives that were positive in the triage test, by age. Odds ratios (ORs) were calculated to examine the risk of histopathological outcomes between arms for the ITS and PP populations. Statistical significance was determined to be $p < 0.05$. The analysis was completed using the statistical package SAS 9.4. This study is reported as per the Consolidated Standards of Reporting Trials (CONSORT) guideline (S2 Text).

## Results

Although the entire population was randomized using a method that produces equal proportion of women by arm and ensures that women cannot change arm (using the PIN), there was

still a larger number of invitations sent in the HPV arm (212,416 women) than in the cytology arm (183,309 women). The mean age among those invited in the cytology arm was 44.5 (median: 44.0 and interquartile range: 36.0 to 51.0), and, in the HPV arm it, was 45.8 (median: 45.0 and interquartile range: 37.0 to 54.0). Age estimates for the ITS population were as follows: The mean in the cytology arm was 44.2 (median: 43 and interquartile range: 36.5 to 50.0), and, in the HPV arm, the ITS mean age was 46.0 (median: 46.0 and interquartile range: 38.0 to 54.0). Age estimates were similar in the PP population (cytology arm was 44.3 (median: 43.0 and interquartile range: 36.0 to 51.0), and the HPV arm was 46.1 (median: 46.0 and interquartile range: 38.0 to 54.0). The imbalance between arms was only found in the number of new invitations (Tables 1 and 2) issued during the first months of the trial. Reminder invitations are sent every year to women who did not attend after being invited last year, and new invitations are sent to women for whom the age-appropriate screening interval (3 or 5 years) had passed since their last screening test was taken. Because of this imbalance, attendance rates should only be compared within invitation category. There was a small, but statistically significant, increase in attendance for women receiving new invitations (but not reminder invitations) in the HPV arm (Tables 1 and 2).

In the PP analyses (women who had participated in the organized screening program), 8.81% (95% confidence interval [CI] 8.65 to 8.98) of women were screen positive in the HPV arm (Table 3), but, after reflex cytology, only 2.26% (95% CI 2.17 to 2.35) of women were referred (Table 1). In the cytology arm, 3.25% (95% CI 3.14 to 3.37) of women were screen positive (Table 4), but, after HPV triage of low-grade cytology, only 1.83% (95% CI 1.75 to 1.92) of women were referred (Table 1). The overall proportion of test-positive baseline results that had a CIN2+ in histology was somewhat lower in the HPV arm 45.80% (95% CI 43.85 to 47.76) versus the cytology arm 50.75% (95% CI 48.35 to 53.15); Tables 3 and 4). The proportion of women positive according to the primary test analysis that were positive in the triage test declined with increasing age, in the HPV and in the cytology arm (Tables 3 and 4). The proportion of referred women who had a biopsy taken was slightly higher in the HPV screening arm (95.04% versus 93.85% in the cytology arm). Both HPV prevalence, abnormal cytology prevalence, and the predictive values of the triaging also declined with increasing age, in both arms (Table 3 and 4).

In the primary analyses of the study (CIN2+ detection rates among women attending organized screening), the total detection rate was somewhat higher in the HPV arm (1.03% (95% CI 0.98 to 1.10) versus 0.93% (95% CI 0.87 to 0.99) in the cytology arm, $p$ for non-inferiority <0.0001; OR 1.11 (95% CI 1.02 to 1.22)) (Table 5). This increase was attributable to an increased detection rate of CIN2 in histopathology (Table 5). The yield of cervical intraepithelial neoplasia grade 3 (CIN3) was the same between arms (Table 5). The point estimate for detection of invasive cancers was somewhat lower in the HPV arm compared to the cytology arm (OR 0.79 (95% CI 0.53 to 1.18)), but formal non-inferiority testing found that both arms were equally effective in detection of invasive cervical cancer ($p < 0.0001$) (Table 5). For estimation of resource use, we chose to count the procedures induced in the 2 arms (Table 5) as the actual price of tests, biopsies, and clinical procedures is variable over time and in different settings. We also performed a sensitivity analysis restricting to women with a new invitation, i.e., excluding women with a reminder invitation. The detection rate for CIN2+ was 747/88,129 women (0.85%, 0.79 to 0.91) in the HPV arm and 551/69,525 women (0.79%, 0.73 to 0.86) in the cytology arm, $p$-value for non-inferiority <0.0001.

In the ITS analysis (all women randomized and assessed for the disease under study, regardless of how the disease was found), the proportions of CIN2+ detected in the 2 arms were similar (HPV arm 1.13% (95% CI 1.07 to 1.19) and cytology arm 1.07% (95% CI 1.01 to 1.14); $p$ for non-inferiority <0.0001) (Fig 1, Table 2)). There were large numbers of cervical

**Table 1. Overall invitations, attendance, and referral to histology by study arm (PP analysis).**

| | Primary HPV policy | | Primary cytology policy | | p-value* |
|---|---|---|---|---|---|
| | n | % (95% CI) | n | % (95% CI) | |
| Overall invitations | 212,416/395,725 | 53.68% (53.52 to 53.83) | 183,309/395,725 | 46.32% (46.17 to 46.48) | <0.0001 |
| Overall attendance* | 110,197/212,416 | 51.88% (51.67 to 52.09) | 90,841/183,309 | 49.56% (49.33 to 49.79) | <0.0001 |
| Attendance by invitation type** | | | | | |
| New invitations | 88,129/137,233 | 64.22% (63.96 to 64.47) | 69,525/109,544 | 63.47% (63.18 to 63.75) | 0.0001 |
| Reminder invitations | 22,067/75,159 | 29.36% (29.04 to 29.69) | 21,302/73,737 | 28.89% (28.56 to 29.22) | 0.0460 |
| Other | 1/24 | 4.17% (0.74 to 20.24) | 14/28 | 50.00% (32.63 to 67.37) | 0.0003 |
| Age among attenders | | | | | |
| 30 to 34 | 16,411/35,174 | 46.66% (46.14 to 47.18) | 16,162/33,875 | 47.71% (47.18 to 48.24) | 0.0057 |
| 35 to 39 | 16,578/32,317 | 51.30% (50.75 to 51.84) | 16,112/31,309 | 51.46% (50.91 to 52.01) | 0.6864 |
| 40 to 44 | 17,728/33,028 | 53.68% (53.14 to 54.21) | 16,688/31,483 | 53.01% (52.45 to 53.46) | 0.0882 |
| 45 to 49 | 16,253/30,564 | 53.18% (52.62 to 53.74) | 15,928/30,188 | 52.76% (52.20 to 53.33) | 0.2997 |
| 50 to 54 | 17,628/32,107 | 54.90% (54.36 to 55.45) | 9,497/21,258 | 44.67% (44.01 to 45.34) | <0.0001 |
| 55 to 59 | 13,484/25,250 | 52.79% (52.79 to 54.02) | 8,780/18,040 | 48.67% (47.94 to 49.40) | <0.0001 |
| 60 to 64 | 12,115/23,976 | 50.53% (49.90 to 51.16) | 7,674/17,156 | 44.73% (43.99 to 45.48) | <0.0001 |
| Referral rate to histology*** | 2,489/110,197 | 2.26% (2.17 to 2.35) | 1,663/90,841 | 1.83% (1.75 to 1.92) | <0.0001 |
| CIN2+ | 1,140/110,197 | 1.03% (0.98 to 1.10) | 844/90,841 | 0.93% (0.87 to 0.99) | 0.0238 |
| CIN3+ | 655/110,197 | 0.59% (0.55 to 0.64) | 524/90,841 | 0.58% (0.53 to 0.63) | 0.7699 |
| Invasive cervical cancer | 46/110,197 | 0.04% (0.03 to 0.06) | 48/90,841 | 0.05% (0.04 to 0.07) | 0.2573 |
| Squamous cell carcinoma | 29/46 | 63.04% (48.60 to 75.48) | 36/48 | 75.00% (61.22 to 85.08) | |
| Adenocarcinoma | 15/46 | 32.61% (20.87 to 47.03) | 11/48 | 22.92% (13.31 to 36.54) | |
| Adenosquamous carcinoma | 1/46 | 2.17% (0.38 to 11.33) | 0/48 | - | |
| Other cervical carcinoma**** | 1/46 | 2.17% (0.38 to 11.33) | 1/48 | 2.08% (0.37 to 10.90) | |

*Chi-squared test of difference between 2 proportions.

**Attendance counted as a cervical sample taken within the organized screening program after invitation, same as overall counts in Fig 1.

***Number of women who have at least 1 histology result during follow-up.

****Unspecified cervical carcinoma.

Note: The PP analysis includes women who were screened and followed up according to the study protocol for that arm (organized screening). Attendance was stratified by invitation type and reported as number and percent that attended.

CI, confidence interval; CIN2+, cervical intraepithelial neoplasia grade 2 or worse; CIN3+, cervical intraepithelial neoplasia grade 3 or worse; HPV, human papillomavirus; PP, per protocol.

biopsies taken both outside of the organized program as well as among screen-negative women (Fig 1). The rate of referral to colposcopy with completed biopsy/histology was 3.89% (95% CI 3.79 to 4.00) of tested women in the HPV arm and 3.53% (95% CI 3.43 to 3.65) of tested women in the cytology arm, measured as the proportion of women who had a histology test result on record during the study follow-up period divided by the number of women who had a cervical HPV test, cytology, or biopsy on record during the same period (Table 2). There were 87 invasive cervical cancers detected in the HPV arm (0.07% (95% CI 0.06 to 0.09) of all participating women) and 91 in the cytology arm (0.09% (95% CI 0.07 to 0.11) of all participating women) (p for non-inferiority <0.0001) (Table 5).

## Discussion

The main outcome of our large-scale RHP was that the HPV screening policy was non-inferior in terms of effectiveness and acceptability. Indeed, in the HPV arm, there was a slightly higher detection rate of CIN2+ following the baseline screening. The HPV policy will also include

**Table 2. Overall invitations, attendance, and referral to histology by study arm (ITS analysis).**

| | Primary HPV policy | | Primary cytology policy | | p-value* |
|---|---|---|---|---|---|
| | *n* | % (95% CI) | *n* | % (95% CI) | |
| Overall invitations | 212,416/395,725 | *53.68% (53.52 to 53.83)* | 183,309/395,725 | *46.32% (46.17 to 46.48)* | <0.0001 |
| Overall attendance | 120,240/212,416 | 56.61% (56.40 to 56.82) | 99,340/183,309 | 54.19% (53.96 to 54.42) | <0.0001 |
| Attendance by invitation type | | | | | |
| New invitations | 94,084/137,233 | 68.56% (68.31 to 68.80) | 74,167/109,544 | 67.71% (67.43 to 67.98) | 0.0002 |
| Reminder invitations | 26,140/75,159 | 34.78% (34.44 to 35.15) | 25,145/73,737 | 34.10% (33.76 to 34.44) | 0.1052 |
| Other | 16/24 | 66.7% (46.4 to 82.2) | 28/28 | 100.0% (89.9 to 100) | 0.0013 |
| Age among attenders | | | | | |
| 30 to 34 | 18,323/35,174 | 52.09% (51.57 to 52.61) | 17,662/33,875 | 52.14% (51.61 to 52.67) | 0.8954 |
| 35 to 39 | 18,134/32,317 | 56.11% (55.57 to 56.65) | 17,583/31,309 | 56.16% (55.61 to 56.71) | 0.8989 |
| 40 to 44 | 19,369/33,028 | 58.64% (58.11 to 59.17) | 18,353/31,483 | 58.29% (57.75 to 58.84) | 0.3672 |
| 45 to 49 | 17,908/30,564 | 58.59% (58.04 to 59.14) | 17,593/30,188 | 58.28% (57.72 to 58.83) | 0.4382 |
| 50 to 54 | 19,262/32,107 | 59.99% (59.46 to 60.53) | 10,702/21,258 | 50.34% (49.67 to 51.02) | <0.0001 |
| 55 to 59 | 14,737/25,250 | 58.36% (57.76 to 58.97) | 9,468/18,040 | 52.48% (51.75 to 53.21) | <0.0001 |
| 60 to 64 | 12,507/23,976 | 52.16% (51.53 to 52.80) | 7,979/17,156 | 46.51% (45.76 to 47.26) | <0.0001 |
| Referral rate to histology** | 4,682/120,240 | 3.89% (3.79 to 4.00) | 3,510/99,340 | 3.53% (3.42 to 3.65) | <0.0001 |
| CIN2+ | 1,359/120,240 | 1.13% (1.07 to 1.19) | 1,062/99,340 | 1.07% (1.01 to 1.13) | 0.1718 |
| CIN3+ | 782/120,240 | 0.65% (60.49 to 69.58) | 669/99,340 | 0.67% (0.62 to 0.73) | 0.5065 |
| Invasive cervical cancer | 87/120,240 | 0.07% (0.06 to 0.09) | 91/99,340 | 0.09% (0.07 to 0.11) | 0.1147 |
| Squamous cell carcinoma | 57/87 | 65.5% (55.1 to 74.9) | 65/91 | 71.4% (61.5 to 80.0) | |
| Adenocarcinoma | 27/87 | 31.0% (22.0 to 41.3) | 24/91 | 26.4% (18.1 to 36.1) | |
| Adenosquamous carcinoma | 1/87 | 1.1% (0.1 to 5.5) | 1/91 | 1.1% (0.1 to 5.3) | |
| Other cervical carcinoma*** | 2/87 | 2.3% (0.4 to 7.4) | 1/91 | 1.1% (0.1 to 5.3) | |

*Chi-squared test of difference between 2 proportions.

**Number of women who have at least 1 histology result during follow-up.

***Clear cell carcinoma, undifferentiated carcinoma, and malignant tumor without further specification.

Note: The ITS analysis includes outcomes for all women who were invited to screening and were randomized to the respective arms, regardless of whether they received the screening test under study. Italicized values are row percentages. Overall attendance in the primary HPV policy arm was 120,240 women, and, in the cytology arm, it was 99,340 women. Overall attendance is stratified by invitation type and reported as number and percent that attended.

CI, confidence interval; CIN2+, cervical intraepithelial neoplasia grade 2 or worse; CIN3+, cervical intraepithelial neoplasia grade 3 or worse; HPV, human papillomavirus; ITS, intention to screen.

retesting of women who are HPV positive but cytology negative 3 years later and referral of women with HPV persistence. It is therefore possible that a further gain in increased protection from HPV-based screening will be seen after referring women with HPV persistence. However, as the baseline yield was already marginally higher (driven largely by CIN2), our data indicate that the HPV screening policy is effective. The slightly higher attendance rate among women who had been issued new invitations also suggests that the policy is acceptable to the population.

As there are no other similarly large, randomized health services studies to the best of our knowledge, direct comparison to others is difficult. However, our results complement the growing body of evidence on primary HPV-based screening from pilot implementations studies and routine practice. Of 5,637 women ages 35 to 60 screened with HPV in the organized routine program of Tampere, Finland, 369 were HPV positive (6.5%), and 54 had cytological abnormalities (detected through conventional cytology)—baseline positivity rates that are slightly lower than ours [15]. Overall detection rates for CIN2+ were significantly higher in

**Table 3. Primary HPV policy screening and histopathology outcomes among 110,197 women who attended PP (organized screening).**

| Age category | HPV positivity, n (%, 95% CI)* | Number and proportion (95% CI) of HPV positives with abnormal reflex cytologies[1] (95% CI)* | - | CIN2+ in histology among women with a histopathology[2], n (%, 95% CI)* |
|---|---|---|---|---|
| 30 to 34 | 2,796/16,411 (17.04%, 16.47 to 17.62) | 823/2,7996 (29.43%, 27.77 to 31.15) | | 404/783 (51.60%, 48.09 to 55.09) |
| 35 to 39 | 1,741/16,578 (10.50%, 10.04 to 10.98) | 518/1,741 (29.75%, 27.64 to 31.93) | | 255/498 (51.20%, 46.81 to 55.58) |
| 40 to 44 | 1,488/17,728 (8.39%, 7.99 to 8.81) | 440/1,488 (29.57%, 27.29 to 31.93) | | 207/425 (48.71%, 43.97 to 53.46) |
| 45 to 49 | 1,223/16,253 (7.52%, 7.13 to 7.94) | 328/1,223 (26.82%, 24.39 to 29.36) | | 119/313 (38.02%, 32.76 to 43.65) |
| 50 to 54 | 1,058/17,628 (6.00%, 5.66 to 6.36) | 242/1,058 (22.87%, 20.42 to 25.48) | | 68/231 (29.44%, 23.83 to 35.56) |
| 55 to 59 | 767/13,484 (5.69%, 5.31 to 6.09) | 142/767 (18.51%, 15.88 to 21.38) | | 47/134 (35.07%, 27.35 to 43.44) |
| 60 to 64 | 639/12,115 (5.27%, 4.89 to 5.68) | 126/639 (19.72%, 16.77 to 22.94) | | 40/105 (38.10%, 29.19 to 47.65) |
| Total | 9,712/110,197 (8.81%, 8.65 to 8.98) | 2,619/9,712 (26.97%, 26.09 to 27.86) | | 1140/2,489 (45.80%, 43.85 to 47.76) |

*Cochran–Armitage test for trend, $p < 0.0001$.

[1]All women with abnormal cytology are referred for colposcopy, but not all women attend and not all women have a biopsy taken (see next column).

[2]Number of unique women with at least 1 cervical histopathology.

CI, confidence interval; CIN2+, cervical intraepithelial neoplasia grade 2 or worse; HPV, human papillomavirus; PP, per protocol.

primary HPV-based screening compared to cytology; however, follow-up included the repeat testing of HPV–positive/cytology-negative women and subsequent referral of women with HPV persistence [15]. Nationwide rollout of primary HPV-based screening was completed in the Netherlands in 2017 in a nonrandomized manner, but, otherwise, the results were very similar to our study [16]. A large, observational study examined the pilot implementation of HPV-based screening in England. The rollout of the new method was done by clusters of general practices and corresponding colposcopy services, comparing contemporaneously to existing cytology practices. The age distribution differed compared to the analysis presented here

**Table 4. Primary cytology policy screening and histopathology outcomes among 90,841 women who attended PP (organized screening).**

| Age category | Abnormal cytology, n (%, 95% CI)* | HSIL- and HPV–positive ASCUS/LSIL[1] | - | CIN2+ in histology among women with a histopathology[2], n (%, 95% CI)* |
|---|---|---|---|---|
| 30 to 34 | 819/16,162 (5.07%, 4.74 to 5.41) | 626 | | 332/596 (55.70%, 51.69 to 59.66) |
| 35 to 39 | 574/16,112 (3.56%, 3.29 to 3.86) | 380 | | 190/360 (52.78%, 47.61 to 57.90) |
| 40 to 44 | 518/16,688 (3.10%, 2.85 to 3.38) | 279 | | 147/270 (54.44%, 48.47 to 60.32) |
| 45 to 49 | 497/15,928 (3.12%, 2.86 to 3.40) | 242 | | 98/228 (42.98%, 36.66 to 49.48) |
| 50 to 54 | 253/9,497 (2.66%, 2.35 to 3.00) | 117 | | 30/104 (28.85%, 20.76 to 38.10) |
| 55 to 59 | 156/8,780 (1.78%, 1.52 to 2.07) | 69 | | 25/61 (40.98%, 29.20 to 53.61) |
| 60 to 64 | 138/7,674 (1.80%, 1.52 to 2.11) | 59 | | 22/44 (50.00%, 35.46 to 64.54) |
| Total | 2,955/90,841 (3.25%, 3.14 to 3.37) | 1,772 | | 844/1,663 (50.75%, 48.35 to 53.15) |

*Cochran–Armitage test for trend, $p < 0.0001$.

[1]All these women were referred for colposcopy, but not all women attend and not all women had a biopsy taken (see next column).

[2]Number of unique women with at least 1 cervical histopathology.

ASCUS, atypical squamous cells of undetermined significance; CI, confidence interval; CIN2+, cervical intraepithelial neoplasia grade 2 or worse; HPV, human papillomavirus; HSIL, high-grade squamous intraepithelial lesions; LSIL, low-grade squamous intraepithelial lesion; PP, per protocol.

**Table 5. Histopathology outcomes and overall detection rates by analysis and study arm.**

| Histopathologically confirmed diagnosis | Primary HPV policy *n* (%) | Primary cytology policy *n* (%) | OR (95% CI) | Non-inferiority test |
|---|---|---|---|---|
| ITS | | | | |
| CIN2 | 840/120,240 (0.70%, 0.6 to −0.75) | 604/99,340 (0.61%, 0.56 to 0.66) | 1.15 (1.04 to 1.28) | <0.0001 |
| CIN3 | 682/120,240 (0.57%, 0.53 to 0.61) | 591/99,340 (0.59%, 0.55 to 0.64) | 0.95 (0.85 to 1.06) | <0.0001 |
| CIN2+* | 1,359/120,240 (1.13%, 1.07 to 1.19) | 1,062/99,340 (1.07%, 1.01 to 1.14) | 1.06 (0.98 to 1.15) | <0.0001 |
| Invasive cervical cancer | 87/120,240 (0.07%, 0.06 to 0.09) | 91/99,340 (0.09%, 0.07 to 0.11) | 0.79 (0.59 to 1.06) | <0.0001 |
| PP | | | | |
| CIN2 | 711/110,197 (0.65%, 0.60 to 0.69) | 484/90,841 (0.53%, 0.49 to 0.58) | 1.21 (1.08 to 1.36) | <0.0001 |
| CIN3 | 596/110,197 (0.54%, 0.50 to 0.59) | 486/90,841 (0.54%, 0.49 to 0.58) | 1.01 (0.90 to 1.14) | <0.0001 |
| CIN2+* | 1,140/110,197 (1.03%, 0.98 to 1.10) | 844/90,841 (0.93%, 0.87 to 0.99) | 1.11 (1.02 to 1.22) | <0.0001 |
| Invasive cervical cancer | 46/110,197 (0.04%, 0.03 to 0.06) | 48/90,841 (0.05%, 0.04 to 0.07) | 0.79 (0.53 to 1.18) | <0.0001 |

*AIS reported only in this summary variable.

AIS, adenocarcinoma in situ; CI, confidence interval; CIN2, cervical intraepithelial neoplasia grade 2; CIN2+, cervical intraepithelial neoplasia grade 2 or worse; CIN3, cervical intraepithelial neoplasia grade 3; HPV, human papillomavirus; ITS, intention to screen; OR, odds ratio; PP, per protocol.

(women ages 24 and above were included) and the direct referral rates were somewhat higher [17]. A recent randomized, non-inferiority trial nested in the Dutch organized cervical screening program compared the clinical accuracy of self-samples with clinician-based sampling for HPV, demonstrating that self-samples taken in population had a similar accuracy, and, therefore, could be used in routine screening [18].

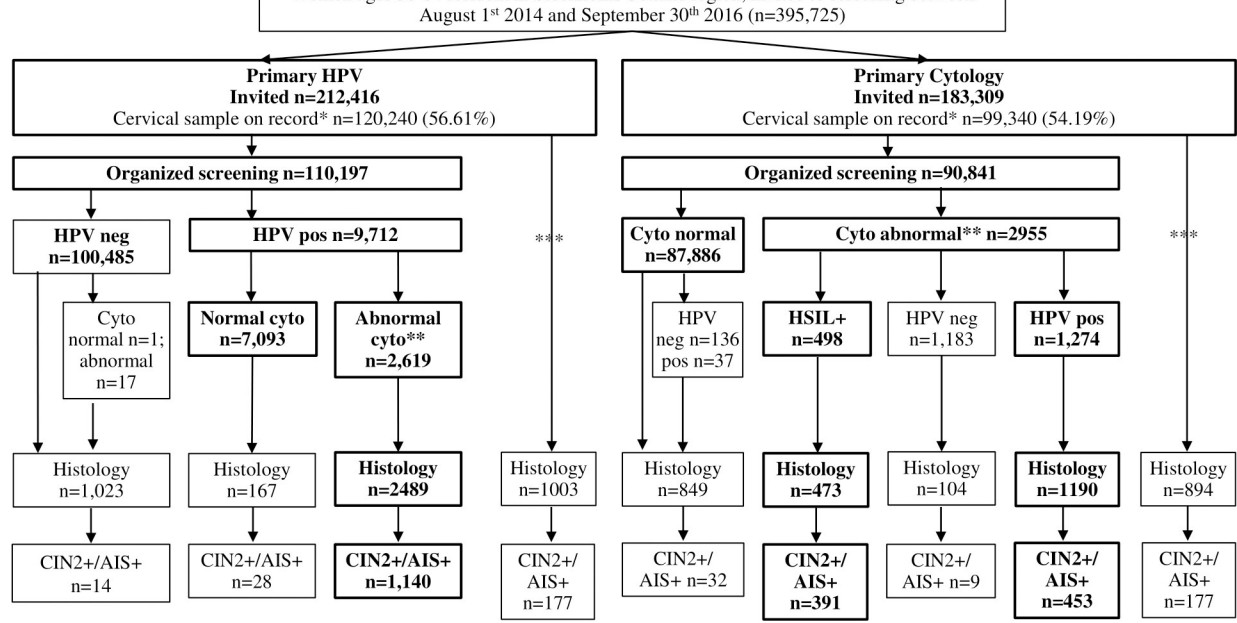

**Fig 1. CONSORT flowchart of study population and outcomes.** * Cervical sample on record: any cervical sample result following the first invitation to screening in the study period. **Abnormal cytology: includes ASCUS/AGC+. ***Includes protocol violations, inadequate samples, pathology samples without proceeding smear, non-organized screening samples, and clinically indicated samples (total: *n* = 10,043 in the primary HPV arm and *n* = 8,499 in the primary cytology arm). Note: Bolded text denotes the PP population participation and outcomes. AGC+, atypical glandular cells or worse; AIS+, adenocarcinoma in situ or worse; ASCUS, atypical squamous cells of undetermined significance; CIN2+, cervical intraepithelial neoplasia grade 2 or worse; CONSORT, Consolidated Standards of Reporting Trials; Cyto, cytology; HPV, human papillomavirus; PP, per protocol.

This analysis presents only the baseline results of the randomized implementation of primary HPV-based screening. However, all screening tests had at least 6 months of follow-up, thus capturing histopathological results of women referred directly for colposcopy. Regarding the significance of trends detected in the results, given the size of the trial, even rather small trends and differences between, e.g., participation, will be highly statistically significant.

There was an imbalance in the number of women sent new invitations. Although the exact reason for this is unknown, the imbalance was restricted to the first months of the study, suggesting that either it is a chance event or may be related to an accumulation of eligible women in the new policy arm while the information technology (IT) system for sending invitations updated. While attendance rates must therefore be analyzed stratified by invitation type, we cannot identify that this imbalance could have resulted in any other major possible source of bias in terms of effectiveness in detecting CIN2+. The possibility that it could have introduced minor biases in unforeseen ways cannot be excluded, however.

As the 2 screening policies compared used the same tests (HPV testing and cytology), referring women who were positive in both the screening test and the triage test, it is somewhat surprising that more women were referred and a higher yield of CIN2+ was found in the arm with HPV testing first. A conceivable explanation is that the greatly reduced number of cytologies (only 9% triage cytologies, as compared to 100% primary screening cytologies) may have resulted in a more cautious interpretation (referring more women than necessary to avoid false negatives). It is also possible that a screening strategy with a more sensitive test first (HPV) followed by a more specific test (cytology) could result in a greater yield of CIN2+. Histopathological diagnosis can be variable, and the fact that the randomization was not blinded could have, conceivably, impacted the histopathological interpretation. However, it should be noted that the purpose of the trial was to examine the whole effect, in real life, when the diagnostic workup following an HPV or a cytology primary screen is not blinded.

The major harm anticipated in the HPV arm was an excess referral rate (largely driven by CIN2) to colposcopy. However, the excess referral rate was only moderately increased (0.36%), and, in a previous publication, we determined that HPV-based screening did not result in overdiagnosis when compared to cytology and examined over 13 years of follow-up [19]. The most effective strategy for avoiding excess referrals among women with HPV positive low-grade lesions is currently under intense investigation, but possibilities could include repeat cytology, risk stratification based on HPV genotype, or molecular triage. It is noteworthy that almost a whole percent of women with normal screening results (HPV negative or cytology normal) had a biopsy taken (with very few CIN2+ lesions detected), which implies that sufficient colposcopy capacity for HPV screening may already exist, as has been found previously [11]. We note that the strategies referring HPV–positive women with a normal cytology result in large referral rates [20]. In the present trial, referral required abnormal cytology among the HPV positives, which is a likely reason as to why we saw only a limited increase in referrals with HPV-based screening. This RHP was carried out in the organized and population-based screening program. The age range mirrored the EU recommendations for primary HPV-based screening, and the follow-up strategies were unchanged compared to routine practices. Therefore, the results are directly generalizable to the screening program and confirm the safety, effectiveness, and acceptability of HPV-based screening. National Swedish guidelines mandating HPV-based screening were issued in 2015 and are similar to the intervention arm of this RHP.

In conclusion, our large-scale real-life RHP of primary HPV-based screening found comparable participation, referral, and detection rates. Real-life evidence of acceptability and effectiveness is important for introducing new policies.

## Notes

K. Miriam Elfström affirms that this manuscript is an honest, accurate, and transparent account of the study being reported; that no important aspects of the study have been omitted; and that any discrepancies from the study as planned have been explained.

## Supporting information

**S1 Text. Study protocol: Randomized implementation of primary HPV testing in the organized screening for cervical cancer in Stockholm.** HPV, human papillomavirus.
(PDF)

**S2 Text. CONSORT 2010 checklist of information to include when reporting a randomized trial.** CONSORT, Consolidated Standards of Reporting Trials.
(DOC)

## Author Contributions

**Conceptualization:** K. Miriam Elfström, Kristina Elfgren, Joakim Dillner.

**Data curation:** Daniel Öhman, Maria Hortlund, Karin Sundström.

**Formal analysis:** K. Miriam Elfström.

**Funding acquisition:** K. Miriam Elfström, Joakim Dillner.

**Investigation:** Carina Eklund, Kristina Elfgren.

**Methodology:** Joakim Dillner.

**Project administration:** Carina Eklund, Helena Lamin.

**Resources:** Joakim Dillner.

**Supervision:** Joakim Dillner.

**Validation:** K. Miriam Elfström.

**Visualization:** K. Miriam Elfström.

**Writing – original draft:** K. Miriam Elfström, Joakim Dillner.

**Writing – review & editing:** K. Miriam Elfström, Karin Sundström, Joakim Dillner.

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
