## [Editor Report · Decision Letter 0]

19 Feb 2021

Dear Dr Dillner, 

Thank you for submitting your manuscript entitled "Effectiveness of organized primary HPV-based cervical screening: randomized healthcare policy" for consideration by PLOS Medicine.

Your manuscript has now been evaluated by the PLOS Medicine editorial staff as well as by an academic editor with relevant expertise and I am writing to let you know that we would like to send your submission out for external peer review.

Kind regards,

Dr Raffaella Bosurgi

Executive Editor 

PLOS Medicine

---

## [Decision Letter · Decision Letter 1]

7 Jun 2021

Dear Dr. Dillner,

Thank you very much for submitting your manuscript "Effectiveness of organized primary HPV-based cervical screening: randomized healthcare policy" (PMEDICINE-D-21-00811R1) for consideration at PLOS Medicine. 

Your paper was reviewed by four independent reviewers, including a statistical reviewer, and has been discussed among all the editors here and with an academic editor with relevant expertise. The reviews are appended at the bottom of this email and any accompanying reviewer attachments can be seen via the link below:

[LINK]

In light of these reviews, I am afraid that we will not be able to accept the manuscript for publication in the journal in its current form, but we would like to consider a revised version that addresses the reviewers' and editors' comments. Obviously we cannot make any decision about publication until we have seen the revised manuscript and your response, and we plan to seek re-review by one or more of the reviewers. 

We expect to receive your revised manuscript by Jun 28 2021 11:59PM. Please email us (plosmedicine@plos.org) if you have any questions or concerns.

We look forward to receiving your revised manuscript. 

Sincerely,

Louise Gaynor-Brook, MBBS PhD

Associate Editor 

PLOS Medicine

plosmedicine.org

Data availability:

PLOS Medicine requires that the de-identified data underlying the specific results in a published article be made available, without restrictions on access, in a public repository or as Supporting Information at the time of article publication, provided it is legal and ethical to do so. Please note that, if the data are only freely available upon request, a study author cannot be the contact person for the data. Please provide an alternative, independent contact for data requests, and their web or email address.

Title: Please revise your title according to PLOS Medicine's style. Please place the study design in the subtitle (ie, after a colon). We suggest “Organized primary human papillomavirus-based cervical screening: a randomized healthcare policy trial” or similar

Abstract:

Please report your abstract according to CONSORT for abstracts, following the PLOS Medicine abstract structure (Background, Methods and Findings, Conclusions) http://www.consort-statement.org/extensions?ContentWidgetId=562 taking into account the extra guidance for non-inferiority trials (https://jamanetwork.com/journals/jama/fullarticle/1487502)

Abstract Background: Provide expand on the background to your study and the context of why the study is important. The final sentence should clearly state the study question.

Abstract Methods and Findings:

Please provide brief demographic details of the study population (e.g. age, ethnicity, etc)

Please provide the number in each group. 

Please include the study design, and the dates during which study enrollment and follow up occurred

In the last sentence of the Abstract Methods and Findings section, please describe 2-3 of the main limitation(s) of the study's methodology.

Please begin your Abstract Conclusions with "In this study, we observed ..." to briefly summarise the main findings of your study. Please address the study implications, mentioning only specific implications substantiated by the results.

Author Summary:

In the final bullet point of ‘What Do These Findings Mean?’, please describe the main limitation(s) of the study in non-technical language.

Introduction:

Please address past research (have there been any other studies reporting on implementation of HPV-based screening within RHPs?) and explain the potential importance of your study. 

Line 66 - please define RCT at first use

Please conclude the Introduction with a clear description of the study question or hypothesis (framed more as a question than ‘we report on ‘).

Methods:

Please include the study protocol document and analysis plan, with any amendments, as Supporting Information to be published with the manuscript if accepted.

Thank you for providing a CONSORT checklist. Please revise this to use section and paragraph numbers, rather than page numbers, and ensure to take into account the extra guidance for non-inferiority trials (https://jamanetwork.com/journals/jama/fullarticle/1487502). 

Please add the following statement, or similar, to the Methods: "This study is reported as per the Consolidated Standards of Reporting Trials (CONSORT) guideline (S1 Checklist)." 

Line 99 - please define ASCUS and LSIL at first use 

Line 140 - please define HSIL at first use 

Line 157 - Please replace "subject" with participant, individual, or person.

Results: 

Please provide a table (Table 1) showing the baseline characteristics of the study population.

The sample size listed in the submitted manuscript and the trial registry differ. Please explain the discrepancy.

Paragraph 2 - please specify what the numbers in brackets represent

Lines 206 / 209 / 218 - please specify the comparison group

Line 210 - please omit ‘both’ 

Line 222 - please clarify in which group ‘detection of invasive cancers was somewhat lower’

Line 231 - please clarify which result corresponds to which group

Discussion:

Please present and organize the Discussion as follows: a short, clear summary of the article's findings; what the study adds to existing research and where and why the results may differ from previous research; strengths and limitations of the study; implications and next steps for research, clinical practice, and/or public policy; one-paragraph conclusion.

Please remove all subheadings within your Discussion e.g. Limitations and other considerations

Line 301 - please correct reference call out in superscript

Line 312 - please temper assertions of primacy by adding ‘to the best of our knowledge’ or similar

Tables:

In each table, when a p value is given, please specify the statistical test used to determine it.

Please define abbreviations used in the table legend of each table.

Please present numerators and denominators for percentages in each table.

References:

Please ensure that journal name abbreviations match those found in the National Center for Biotechnology Information (NCBI) databases, and are appropriately formatted and capitalised e.g. refs 7, 11, etc. Please also see https://journals.plos.org/plosmedicine/s/submission-guidelines#loc-references for further details on reference formatting.

Comments from the reviewers:

Reviewer #1: I confine my remarks to statistical aspects of this paper. These were very well done (and clearly explained) and I have only a couple of very minor requests before I can recommend publication.

p. 5 Line 104-105 Please give a citation for this and, perhaps, some reasoning behind it. It seems counter-intuitive.

p. 8 Line 163-164 Please give a citation for the Farrington-Manning score, and describe it, as it is relatively little known.

Peter Flom

Reviewer #2: The present manuscript by Elfström et al. deals with a topic that most experts in public health would agree is very important, namely, supplementing studies of efficacy with studies of effectiveness before implementing or significantly modifying interventional programme affecting large numbers of people.

The present authors have conducted a landmark study in which evidence-based modifications to improve a long-running, population-based cervical cancer screening programme have been evaluated in a trial involving randomization of all of the over 200,000 women invited to attend the programme over a 4-year period.

The results of the presently reported registered randomized public health policy trial verify that under real-life conditions the programme fulfilled the expectations of policymakers in modifying the programme.

This achievement is likely to receive wide attention by professionals and policymakers in public health and in women's health. The paper not only provides a model for future efforts to implement and modify large scale programmes. 

The paper is also highly relevant, even though HPV vaccination in younger women has been universally recommended in the past decade for primary prevention of cervical cancer - because for older women who have already been infected by carcinogenic HPV organized cervical cancer screening will continue to be the most effective option for cervical cancer control for many years. Moreover, for younger women screening programmes will need to be adapted to monitor and evaluate vaccination programmes, and the landmark study of Elfström et al. provides a valuable example of how such modifications can be evaluated.

In addition to the topic, the methodology of the study, the organisation of the paper and the presentation and discussion of the results are exemplary and are likely to be emulated by researchers in the coming years.

I therefore recommend the paper for publication without modification. 

Reviewer #3: The manuscript is much clearer now. I have no mayor comments. I have spotted two very minor typos:

Page 21 line 206, there is an extra parenthesis '(45.80%'

Table 4. There is a 2 superscript after CIN2+ which does not have a footnote.

Reviewer #4: The authors describe: Effectiveness of organized primary HPV-based cervical screening: randomized healthcare policy.

This is indeed a large randomized study in comparing HPV based cervical cancer screening with cytology based screening.

However several issues need to be addressed in my opinion.

1. It is not clear to me whether women had to consent to this study, and if they were aware of randomisation and different assessment policies in the lab?? If no consent was obtained, explain why this was left out of the procedure.

2. Randomisation was done on the last digit of the PIN. But a large difference between size of the groups as well as age and attendence was found. Was this difference also there at time of invitation and if so why was it not corrected during the study to create equal groups at least at invitation.

3. the groups consist of women with new as well as repeat invitations, and also on this aspect the groups differ. I would prefer to have groups as homogeneous as possible and would like to see an analysis of the new invitations only as especially repeat invitations of non responders may cause more bias.

4. The analysis is done with data gathered in 2017. Given the 3 year interval for recall in case of HPV pos cytology negative or Low grade cytology HPV neg I would like to have those data incorporated as it will provide a much more rliable picture. We expect especially in the HPV positive cytology negative group a substantial number of CIN 2+, as other studies show that HPV screening with a single cytology triage misses a substantial number of cases.

5. Biopsies are used as histology result in this study. The inter and even intra observer variability is quite large especially on biopsies. this needs to be commented on in the discussion.

6. The referrel rate went up with an extra 0.36% in the HPV arm. This is much less as experienced in other countries. For instance in the netherlands referrals more than doubled after introduction of HPV based screening. On the other hand a 3.5% referral rate in the cytology arm is high compared to what we are used to, especially since the second triage step is not included here. Please comment on this.

7. The most recent refference is of early 2019. Several important studies on this subject have been published since then, for instance Polman et al. Lancet Oncology, 20(2), 229-238. https://doi.org/10.1016/S1470-2045(18)30763-0, also a RCT in a screening population. so references must be updated and incorporated in the discussion.

[LINK]

---

## [Decision Letter · Decision Letter 2]

21 Jul 2021

Dear Dr. Dillner,

Thank you very much for re-submitting your manuscript "Organized primary human papillomavirus-based cervical screening: a randomized healthcare policy trial" (PMEDICINE-D-21-00811R2) for review by PLOS Medicine.

I have discussed the paper with my colleagues and the academic editor and it was also seen again by one reviewer. I am pleased to say that provided the remaining editorial and production issues are dealt with we are planning to accept the paper for publication in the journal.

[LINK]

We look forward to receiving the revised manuscript by Jul 28 2021 11:59PM.   

Sincerely,

Louise Gaynor-Brook, MBBS PhD

Associate Editor 

PLOS Medicine

plosmedicine.org

Requests from Editors:

Throughout the paper, please adapt reference call-outs to the following style: "... low-grade cervical cytological abnormalities [6,7]" (noting the absence of spaces within the square brackets).

Please revise your Author Summary, which should include 2-3 bullet points per subheading, in accessible and non-technical language. Please see e.g. https://journals.plos.org/plosmedicine/article?id=10.1371/journal.pmed.1003686 for a recent example. In the final bullet point of ‘What Do These Findings Mean?’, please describe the main limitation(s) of the study.

Introduction:

Line 110 - please revise beginning of sentence beginning ‘The purpose being…’

Methods:

Please refer to your study protocol / analysis plan early in the Methods section. 

Results:

Line 258 - please clarify ‘The proportion of positives that were positive…’

Please add a final one-paragraph conclusion to your Discussion

References:

Please ensure that journal name abbreviations match those found in the National Center for Biotechnology Information (NCBI) databases, and are appropriately formatted and capitalised.

Please also see https://journals.plos.org/plosmedicine/s/submission-guidelines#loc-references for further details on reference formatting. E.g. Ref 7 should be Contemp Clin Trials, ref 11 should be BMJ Open

There appears to be additional unnecessary information in ref 17

Comments from Reviewers:

Reviewer #4: the authors have addressed all issues raised very well. I recommend acceptance in its present form

[LINK]

---

## [Editor Report · Decision Letter 3]

29 Jul 2021

Dear Dr. Dillner,

Thank you very much for re-submitting your manuscript "Organized primary human papillomavirus-based cervical screening: a randomized healthcare policy trial" (PMEDICINE-D-21-00811R3) for review by PLOS Medicine.

I am afraid that there are remaining issues with your Author Summary, which will need to be resolved before we are able to accept the paper for publication in the journal. Please see more details at the end of this email. 

We look forward to receiving the revised manuscript by Aug 05 2021 11:59PM.   

Sincerely,

Louise Gaynor-Brook, MBBS PhD

Associate Editor 

PLOS Medicine

plosmedicine.org

Requests from Editors:

Please revise the Author Summary to contain 2-3 bullet points per heading to summarise your study in non-technical language. The final bullet point of the final section should indicate the key limitations of the study. We would suggest the following; please ensure that this is an accurate summary of your study before resubmitting to us.

Why was this study done?

- Evidence from clinical trials has shown that primary HPV-based screening results in greater protection against cervical cancer than cytology-based screening. 

- Evidence from real-life implementation of primary HPV-based screening in routine healthcare is scarce.

- We sought to generate evidence on the effectiveness and acceptability of HPV-based screening within an organized screening program. 

What did the researchers do and find? 

- Women invited to organized cervical screening in Stockholm, Sweden were randomized to receive either primary HPV-based screening or primary cytology-based screening.

- Women were followed for histopathological confirmed high grade cervical cancer precursors (CIN2+) as the primary outcome. Screening attendance and referral rates to histology were secondary outcomes. 

- The detection of CIN2+ was comparable between both screening methods. 

- Among women receiving primary HPV-based screening, there was a modestly increased attendance after new invitations and an increased rate of referral with completed biopsy. 

What do these findings mean? 

- When organised primary HPV-based screening was implemented in a real-life screening program, it was both acceptable and effective. 

- The main limitations of this analysis are that only baseline results are presented and there was an imbalance in invitations between the two screening methods. 

[LINK]

---

## [Editor Report · Decision Letter 4]

2 Aug 2021

Dear Dr Dillner, 

On behalf of my colleagues and the Academic Editor, Dr. James Brenton, I am pleased to inform you that we have agreed to publish your manuscript "Organized primary human papillomavirus-based cervical screening: a randomized healthcare policy trial" (PMEDICINE-D-21-00811R4) in PLOS Medicine.

PRESS

Sincerely, 

Louise Gaynor-Brook, MBBS PhD 

Associate Editor 

PLOS Medicine